# Minimum Infective Dose of a Lumpy Skin Disease Virus Field Strain from North Macedonia

**DOI:** 10.3390/v12070768

**Published:** 2020-07-16

**Authors:** Janika Wolff, Kiril Krstevski, Martin Beer, Bernd Hoffmann

**Affiliations:** 1Institute of Diagnostic Virology, Friedrich-Loeffler-Institut, Federal Research Institute for Animal Health, Südufer 10, D-17493 Greifswald-Insel Riems, Germany; janika.wolff@fli.de (J.W.); Martin.Beer@fli.de (M.B.); 2Faculty of Veterinary Medicine, University “Ss. Cyril and Methodius” Skopje, Lazar Pop Trajkov 5–7, 1000 Skopje, North Macedonia; krstevski@fvm.ukim.edu.mk

**Keywords:** Capripox, Lumpy Skin Disease, LSDV, minimum infective dose, experimental inoculation, field strain, challenge model

## Abstract

Infection with *Lumpy Skin Disease virus* (LSDV), as well as infections with other Capripox virus species, are described as the most severe pox diseases of production animals and are therefore listed as notifiable diseases under the guidelines of the World Organization for Animal Health (OIE). To our knowledge there is only a single study examining dose dependency, clinical course, viremia, virus shedding, as well as serological response following experimental LSDV “Neethling” inoculation. Here, we inoculated cattle with four different doses of LSDV strain “Macedonia2016”, a recently characterized virulent LSDV field strain, and examined clinical symptoms, viremia, viral shedding, and seroconversion. Interestingly, around 400 cell culture infectious dose_50_ (CCID_50_) of LSDV-“Macedonia2016” were sufficient to induce generalized Lumpy Skin Disease (LSD) in two out of six cattle but with a different incubation time, whereas the other animals of this group showed only a mild course of LSD. However, differences in incubation time, viral loads, serology, and in the clinical scoring could not be observed in the other three groups. In summary, we concluded that experimental LSDV infection of cattle with an infectious virus titer of 10^5^ to 10^6^ CCID_50_/_mL_ of “Macedonia2016” provides a robust and sufficient challenge model for future studies.

## 1. Introduction

*Lumpy Skin Disease virus* (LSDV) together with *Sheeppox virus* (SPPV) and *Goatpox virus* (GTPV) belongs to the genus Capripoxvirus within the family Poxviridae [1]. Since Capripox virus-induced diseases are described as the most severe pox diseases of production animals [2,3,4], they are listed as notifiable diseases under World Organization for Animal Health (OIE) guidelines [5]. SPPV and GTPV infect either sheep or goats, respectively, and some isolates are able to induce clinical disease in both sheep and goats. In contrast, LSDV is, with few exceptions, only found in cattle, buffalos, and other wild ruminants [2,6,7,8,9]. Clinical reactions range from subclinical through mild or moderate to acute [2], with 10 to 50% subclinical infections [10]. After an incubation period of 7–14 days following experimental infection [11,12], and 1–4 weeks in natural outbreaks [7,13], Lumpy Skin Disease (LSD) infections often start with an initial period of fever [12,14]. In the following days, clinical symptoms, like enlarged lymph nodes [14], excessive salivation, and nasal discharge [7,15], as well, as emaciation [15,16,17,18], are observed. Furthermore, the affected cattle develop characteristic skin nodules, which may occur sporadically or even generalized and can cover the whole body [12,14,15,18]. Additionally, these lesions may develop in the gastrointestinal [19], as well, as in the respiratory tract, especially in severely affected animals [18,19]. Transmission of LSDV is described to occur mainly mechanically via blood-feeding insects [20,21,22,23].

The minimum infective dose of almost all LSDV isolates seems to be undefined. There are overall only very few studies describing experimental infection of cattle with LSDV. Most of them used an infective virus dose of around 10^5^ tissue culture infectious dose_50_ (TCID_50_) per mL, and inoculated 0.5 mL [11] or 2 mL [14,24,25,26] per animal. In addition, two other studies inoculated cattle 6× with 1 mL and 4× with 2 mL virus suspension, respectively [21,22]. Intravenous [11,14,24,25] or intradermal/subcutaneous [21,22,26] infection routes were used most often, whereas for one study, 1 mL of fresh semen was spiked with 1 mL of infectious LSDV and inseminated into heifers [27]. For these studies, different virus strains were used: the South African isolate “V248/93” [14,24,27], a South African “Neethling strain” [11,12,20], an LSDV strain from Ismailiya, Egypt [21,22], LSDV strain “Dagestan/2015” from Russia [25], and an LSDV isolate from the Republic of North Ossetia-Alania in 2015 [26].

However, to our knowledge, there are only two studies dealing with different infectious titers of the inoculation material and comparing different inoculation routes with a virulent South African Neethling strain [12,20]. In one study, the intradermal route turned out to be less effective compared to the intravenous route, since it was associated with the development of a local skin nodule and only few animals developed generalized LSD. Nevertheless, the intravenous route is described as highly likely to produce generalized LSDV infections [20]. In the second study, a dose dependency was reported for the minimum infective dose and the timeline for the development and the severity of local lesions in cattle with non-generalized LSD. However, no correlation between the infection dose and the overall severity of the disease or the development of the generalized LSD form could be observed. For the minimum infective dose, the authors also reported a role of the transmission route: whereas 10^1^ TCID_50_/_mL_, inoculated intradermally, was able to establish LSDV infection, and 10^2^ TCID_50_/_mL_ were necessary to produce generalized infections via this route, virus titers of 10^3^ TCID_50_/_mL_ and 10^3.3^ TCID_50_/_mL_ were needed to induce generalized infections via the intranasal or the intravenous route [12].

In our study, we examined the minimum infective dose of the virulent LSDV-“Macedonia2016” field strain, which has already been in vivo characterized. In the preceding study, a virus suspension with a titer of 10^7.4^ TCID_50_/_mL_ was inoculated intravenously (3 mL) and subcutaneously (1 mL) into cattle [28]. In order to evaluate the dose response and the minimum infective dose of LSDV-“Macedonia2016”, we inoculated cattle (*n* = 6 per group) with different virus dosages (from 10^2^ cell culture infectious dose_50_ (CCID)_50_/_mL_ to 10^7^ CCID_50_/_mL_), and analyzed, e.g., clinical course, viremia, virus shedding, and serological response.

## 2. Materials and Methods

### 2.1. Animals

Twenty-four 4- to 6-month-old Holstein-Friesian cattle were housed in the facilities of the Friedrich-Loeffler-Institut—Insel Riems, Germany, under Biosafety level 4 (animal) conditions. All respective animal protocols were reviewed by a state ethics commission and were approved by the competent authority (State Office for Agriculture, Food Safety and Fisheries of Mecklenburg-Vorpommern, Rostock, Germany; Ref. No. LALLF M-V/TSD/7221.3-1-045/18, date of approval 12.09.2018). All animals were in good health, and no symptoms indicating acute infections with other pathogens could be observed.

### 2.2. Experimental Design and Sample Collection

The animals were split in four groups of six cattle each, and four different infection doses were tested (10^2^ cell culture infectious dose_50_ (CCID_50_), 10^4^ CCID_50_, 10^6^ CCID_50_, and 10^7^ CCID_50_). Inoculation was performed intravenously with a *Lumpy Skin Disease virus* (LSDV) field strain isolated from affected cattle during an outbreak in North Macedonia in 2016 [28]. Preparation of virus stock was performed as described in the following: LSDV-“Macedonia2016” was propagated on Madin-Darby bovine kidney (MDBK) cells (Friedrich-Loeffler-Insitut (FLI) internal cell culture collection number CCLV-RIE0261 (CCLV = Collection of Cell Lines in Veterinary Medicine)). Therefore, MDBK cells with a confluence of approx. 90% (achieved approximately 24 h after seeding of cells, seeded in T75 cell culture flasks) were infected using 100 µL virus suspension/T75. After incubation for 7 days at 37 °C, cells were frozen at −80 °C and stored until the start of the animal trial. Each animal received 6 mL of virus suspension diluted in serum-free medium to exclude a possible impact of the inoculation volume regarding, e.g., local reaction at the inoculation site. Back-titration of the used inoculation material revealed the following titers on MDBK cells:Group A: 10^1.8^ CCID_50_/_mL_ ≥ 3.8 × 10^2^ CCID_50_/cattleGroup B: 10^3.6^ CCID_50_/_mL_ ≥ 2.4 × 10^4^ CCID_50_/cattleGroup C: 10^5.6^ CCID_50_/_mL_ ≥ 3.3 × 10^6^ CCID_50_/cattleGroup D: 10^6.8^ CCID_50_/_mL_ ≥ 3.8 × 10^7^ CCID_50_/cattle

Body temperatures were measured daily from 4 days post-infection (dpi) until 21 dpi (normal range of body temperature prior experimental infection: 38.5–39.5 °C). For examination of clinical reactions from 0 dpi to 21 dpi, we modified the clinical score system of Carn and Kitching [12] with regard to our experiences from studying experimental LSDV infection in cattle (Table 1). When reaching a clinical score of ≥10 (human endpoint) or showing generalized LSDV over a period of several days without clear clinical improvement, the respective cattle was euthanized due to ethical reasons. All cattle were sampled at certain time points during the animal trial: −1, 3, 5, 7, 10, 12, 14, 17, 21, and 28 dpi. Thereby, Ethylenediaminetetraacetic acid (EDTA) blood and nasal swabs were taken to examine cell-associated viremia and virus shedding, respectively. Serum samples were used for the analysis of cell-free viremia and serological evaluation. Furthermore, a defined panel of organs was taken during necropsy.

### 2.3. Molecular Diagnostics

First, all organ samples were homogenized in serum-free medium using the TissueLyser II tissue homogenizer (QIAGEN, Hilden, Germany). DNA extraction from all samples taken during the animal trial (EDTA blood, serum, nasal swabs), as well, as the homogenized organ samples was performed with 100 mL sample material using the NucleoMag Vet kit (Macherey-Nagel, Düren, Germany) according to the manufacturer’s instructions on the KingFisher Flex System (Thermo Scientific, Darmstadt, Germany). The extracted nucleic acid was eluated in 100 µL elution buffer. For control of successful DNA extraction, an internal control DNA (IC-2 DNA) was added during the extraction process [29]. Afterwards, a pan Capripox real-time qPCR using the PerfeCTa qPCR ToughMix (Quanta BioSciences, Gaithersburg, MD, USA) was performed according to the protocols of Bowden et al. (primer) [3] and Dietze et al. (probe) [30] using 2.5 µL template DNA in a 12.5 µL qPCR total volume.

### 2.4. Serological Examination

For serological analyses, the serum neutralization test (SNT) and an Enzyme-Linked Immunosorbent Assay (ELISA), the ID Screen Capripox Double Antigen (DA) ELISA (ID.vet, Montepellier, France), were used.

The DA ELISA was performed following the manufacturer’s instructions.

Detection of neutralizing antibodies was performed using an LSDV-specific SNT. Here, serum samples were inactivated for 2 h at 56 °C, followed by preparation of triplicates of log_2_ dilution series in serum-free medium starting from 1:10 (96-well plate format was used). The diluted serum samples were incubated for 2 h at 37 °C and 5% CO_2_ with 50 µL of LSDV-Neethling vaccine strain, which had a titer of 10^3.3^ CCID_50_/mL. Afterwards, MDBK cells (FLI cell culture collection number CCLV-RIE0261) with a concentration of approximately 30,000 cells/100 µL were added to the samples and incubation took place for 7 days at 37 °C and 5% CO_2_. For analyses of the SNT, a light microscope (Nikon Eclipse TS-100 (Nikon, Amsterdam, The Netherlands)) was used. For titer determination, the Spearman-Kärber method was applied [31,32].

## 3. Results

### 3.1. Clinical Scoring

In group A (10^2^ cell culture infectious dose_50_ (CCID_50_)/mL), two animals showed increased body temperature a few days after experimental infection. For animal R/280, a higher body temperature could be detected on 8 days post infection (dpi) and 9 dpi, whereas animal R/203 showed two peaks of increased body temperature, the first at 3 dpi to 4 dpi and the second lasting from 7 dpi to 9 dpi. Furthermore, an increase of body temperature for animal R/824 could be observed at 21 dpi. All other cattle remained unremarkable until the end of the study (data not shown).

For group B (10^4^ CCID_50_/mL), group C (10^6^ CCID_50_/_mL_), and group D (10^7^ CCID_50_/mL), a similar course of body temperatures could be observed. Body temperature of most animals started to increase around 4 dpi/5 dpi, and fever lasted around six to nine days. One animal (R/984), in addition, had a fever peak at 2 and 3 dpi before onset of a longer lasting fever period at 8 dpi. Solely, two animals (R/891 and R/857 of group B) did not show an increased body temperature after experimental infection (data not shown).

In addition to the body temperature, appearance of symptoms typical for LSDV infections as well as the overall clinical score for each cattle (Table 1) was determined daily from 0 until 21 dpi (Figure 1, Figure 2 and Figure 3). Animals with a moderate to severe clinical course had enlarged lymph nodes and showed oral and nasal discharge (Figure 2A), as well, as characteristic pox-like skin lesions (Figure 2B–D), which occurred sporadically or were generalized in some animals.

The observed clinical reactions differed between the four infection groups but also showed some similarities. Whereas the severity of the clinical score varied between and within the groups, onset of clinical symptoms was comparable amongst all groups. In detail, first clinical signs could be seen at 7 dpi in at least one animal of each of the four groups (Figure 1, Figure 3).

In group A, three out of six cattle (R/278, R/279, R/203) showed a mild clinical course with a clinical score not higher than four. Additionally, one animal (R/915) did not show any reaction at all. However, two animals developed severe Lumpy Skin Disease (LSD), but with differences concerning the time point. For R/280, clinical symptoms were observed from 8 dpi on, which became severe with reaching a clinical score of ten, leading to euthanasia at 13 dpi. Animal R/824, which did not show any clinical signs until 18 dpi, surprisingly developed severe symptoms characteristic for *Lumpy Skin Disease virus* (LSDV) infections from 19 dpi on. With a clinical reaction score of seven at 21 dpi, this animal was removed due to ethical reasons (Figure 1A, Figure 3).

In groups B, C, and D, all animals showed clinical reactions following LSDV infection, but with variable severity.

In group B, three out of six cattle (R/857, R/891, R/990) showed mild to moderate clinical disease. A consistent clinical score of one could be observed for R/891, which showed only slight nasal discharge and slightly reduced food intake during a few days post inoculation (9 dpi until 12 dpi and 14 dpi). Similarly, also R/857 showed only mild nasal discharge, slightly reduced food intake, and reduced activity, leading to a clinical reaction score of one to three after inoculation. A moderate clinical course could be observed for R/990, which reached a maximum clinical score of five at 14 dpi, but recovered from LSD completely, and no clinical signs could be observed after 19 dpi. The other three cattle of group B were severely affected, leading to euthanasia of R/888 (clinical score: ten) on 11 dpi and of R/829 (clinical score: seven), and R/984 (clinical score: nine) at 13 dpi (Figure 1B, Figure 3).

Compared to group B, the animals of group C showed a more severe clinical reaction after experimental infection. In this group, two cattle (R/284 and R/921) showed a moderate clinical course reaching a maximum clinical score of six (R/284, 10 dpi and 11 dpi) and five (R/921 10 dpi), and both recovered completely from LSD until the end of the study. However, the remaining four animals of group C showed severe clinical symptoms and had to be euthanized at 11 dpi (R/283, clinical score: nine; R/841, clinical score: eight; R/989, clinical score: nine) and 13 dpi (R/277, clinical score: eight), respectively (Figure 1C, Figure 3).

For group D, a similar clinical course compared to group B could be observed. Here, four out of six cattle showed moderate clinical signs after inoculation with LSDV, leading to maximum clinical scores of four (R/276, R/893) and five (R/282, R/860), respectively. However, R/981, as well as R/988, showed a clinical reaction score of eleven at 11 dpi (clinical score at 10 dpi: R/981: nine; R/988: seven) and were removed from the trial due to ethical reasons (Figure 1D, Figure 3).

### 3.2. Virus Replication and Shedding

For analyses of cell-associated and cell-free viremia, as well as viral shedding, EDTA blood, serum samples, and nasal swabs were taken at different time points during the animal trial and were examined using a pan Capripox real-time qPCR (Table 2, Figure 3, genome copies/µL template Supplemental Appendix A).

In groups A and B, the first positive qPCR results were found at 7 dpi (Group A: R/280_serum, Cq 37.1; Group B: R/984_EDTA blood, Cq 37.4), whereas in group C (R_841_EDTA blood, Cq 34.8) and D (R/282_EDTA blood, Cq 36.5; R/981_EDTA blood, Cq 31.7; R/988_EDTA blood, Cq 33.5 and _serum, Cq 37.5) the first positive samples were detected at 5 dpi. The first positive nasal swabs were found at 12 dpi for groups A (R/280, Cq 34.8; R/824, Cq 37.3) and B (R/857, Cq 38.3; R/891, Cq 37.7; R/984, Cq 31.2), at 10 dpi for group C (R/283, Cq 35.0; R/989, Cq 33.7), and at 7 dpi for group D (R/988, Cq 36.2) (Table 2). Overall, all four groups showed similar patterns of viremia and viral shedding in nasal swabs (Figure 3).

Interestingly, in group A, five out of six animals were tested positive in EDTA blood or serum at least once during the animal trial, and, for four of them, viral shedding also could be observed in at least one nasal swab sample. The only exception is animal R/203, which was negative in all three matrices at all tested time points. However, the observed Cq values are relatively high (29.1–38.6), indicating low virus loads in the samples obtained from cattle of group A. Only these two animals, which had to be euthanized before the end of the trial (R/280, R/824), tested positive in all three matrices before euthanasia.

In group B, again, five out of six cattle showed viremia in either EDTA blood or serum or both at least at two time points during the study. Two of them (R/888, R/829) remained negative in nasal swabs, while the other four animals showed positive results for the nasal swabs at different days post infection. In contrast to group A, only one out of three animals that were euthanized before the end of the study (R/984) tested positive for Capripox virus genome in all three matrices. The other two cattle (R/888, R/829) were negative in nasal swabs during the whole observation period (Table 2 Group B). Interestingly, for animal R/857, no viremia could be detected, but viral genome could be observed in nasal swab samples obtained at 12 and 17 dpi.

Four out of the six animals of group C had to be removed from the trial due to ethical reasons. Three of them (R/277, R/283, R/989) displayed positive results in the pan Capripox real-time qPCR in all three tested matrices. Solely, R/841 showed only viremia, and no viral shedding could be detected by analyzing the nasal swabs. For both surviving animals of this group (R/284, R/921), viremia could be observed for 14 and 11 days, respectively. However, a positive nasal swab was only detected at 12 dpi for R/921. At this day, R/921 also displayed positive results in EDTA blood, serum, and nasal swab (Table 2 Group C, Figure 3) but developed only a moderate clinical course of LSD (Figure 1C, Figure 3).

In general, lower Cq values for the nasal swabs could be observed in two animals of group D (23.2 (R/988, 11 dpi), 24.6 (R/981, 11 dpi), and 26.8 (R/981, 10 dpi) to 27.6 (R/988 10 dpi)) (Table 2 group D) compared to the cattle of the other three groups (lowest: group C, R/989, 11 dpi, Cq 28.5) (Table 2 group A–C). Both, R/981 and R/988, additionally displayed strong viremia and had to be euthanized at 11 dpi due to ethical reasons. The remaining four animals of group D showed only slight to moderate viral genome loads in both EDTA blood and serum, as well as mild to moderate viral shedding. For animal R/860, no viral shedding could be observed by analyses of the nasal swabs (Table 2, group D).

### 3.3. Viral Genome Loads in Certain Organ Samples

During necropsy, a panel of different organ samples was taken and analyzed by pan Capripox real-time qPCR in order to determine the viral genome load (Table 3, genome copies/µL template Supplemental Appendix A). Here, it has to be highlighted that, in ten out of the eleven animals which had to be euthanized before the end of the trial, the cervical lymph node tested positive (Cq value between 25.1 (R/988) and 36.4 (R/829)). The only exception is the cervical lymph node of R/277, which tested negative for Capripox virus genomes. However, all surviving animals except R/915 (Cq 36.7) displayed negative qPCR results for the cervical lymph node. It is noticeable that, in many animals that showed positive results for Capripox virus genomes in at least one organ sample, the cervical lymph node was also tested positive. However, some exceptions could be observed. R/284, R/276, and R/893 displayed pox-like lesions in the lung that tested positive, R/860 was only positive in the lung sample, and R/277 showed positive qPCR results in pox-like lesions of the lung, the lung itself, and skin nodules. Overall, the mediastinal and mesenterial lymph nodes, the spleen, and the liver were tested positive for Capripox virus genome only sporadically. Interestingly, only one animal (R/988) showed positive results in all tested organ samples (Table 3). Taken together, these data indicate that the cervical lymph node, as well as the lung, can be considered a sensitive and suitable organ sample for confirmation of Capripox virus infections.

### 3.4. Serological Response

For serology, the serum neutralization test (SNT), as well as a Double Antigen (DA) ELISA (ID.vet), were performed. At −1 dpi, all animals displayed negative results in the DA ELISA, representing absence of antibodies against Capripox viruses (Table 4). Together with the fact that there have never been cases of Capripox viruses in Germany, all cattle included in the present study can be considered as Capripox-negative. Serological analyses revealed clear differences between group A and the other three groups. Moreover, group B also differs slightly from group C and D, which showed a similar serological pattern (Table 4, Figure 3).

Although two animals of group A developed severe LSD, all animals of group A tested negative in the ELISA, as well as in the SNT (Table 4, Figure 3).

In group B, two (R/891, R/990) out of the three surviving animals tested positive in the SNT starting at 28 dpi, but with relatively low neutralizing titers (R/990 = 1:20; R/891 = 1:40). Whereas R/891 additionally scored clearly positive in the ELISA at 28 dpi and the day of necropsy, R/990 remained negative in this assay. For the third animal (R/857), no antibodies were detectable using both the SNT and the ELISA (Table 4, Figure 3).

Group C and D were comparable regarding the serological response of the surviving cattle. All of them developed antibodies detectable by both methods. Five out of the six remaining animals of groups C and D displayed positive results in the ELISA from 28 dpi on. R/860 already scored positive in the ELISA at day 17 pi. Positive results in the SNT could be observed starting at different time points: 17 dpi (group D: R/860, R/893), 21 dpi (group C: R/921; group D: R/282), and 28 dpi (group C: R/284; group D: R/276) with similar titers. Solely the serum of R/860 (group D) scored markedly higher in the SNT (titer 1:512) than those of the other animals (Table 4).

In all cattle that were euthanized due to ethical reasons before the end of the study, no antibodies could be detected using the SNT and the ELISA (Table 4, Figure 3), which might be mainly due to the early time point of euthanasia.

## 4. Discussion

Since the intravenous inoculation route previously had proved to be highly effective and very likely produced generalized *Lumpy Skin Disease virus* (LSDV) infections [20], the cattle in our study were inoculated with 6 mL of virus suspension intravenously. Furthermore, Carn and Kitching postulated, that a virus titer of 10^3.3^ tissue culture infectious dose_50_ (TCID_50_)_/_mL of a virulent South African LSDV isolate is needed to produce generalized Lumpy Skin Disease (LSD) [12]. In addition, our previous animal trial comparing LSDV vaccine strain “Neethling” and LSDV-“Macedonia2016” also showed that 4 mL of a virus suspension of LSDV-“Macedonia2016” with a titer of 10^7.4^ TCID_50_/_mL_ were able to infect cattle with LSDV and to produce generalized LSD when inoculated intravenously and subcutaneously [28]. Therefore, we decided to use the following approximate infective doses for the examination of dose response and minimum infective dose: 10^2^ cell culture infectious dose_50_ (CCID_50_), 10^4^ CCID_50_, 10^6^ CCID_50_, and 10^7^ CCID_50_.

Interestingly, clinical symptoms displaying mild to generalized forms of LSD could be observed in all four groups. The incubation time was comparable between the four groups, and first clinical signs could be detected at 7 days post infection (dpi) (Figure 1, Figure 3). These findings correlate with the data of several other studies, reporting incubation periods of 4–8 days [11,14,25,26] and 7–14 days [12] after experimental infection, respectively. Nevertheless, two animals of group A (10^2^ CCID_50_) showed no reaction at all (R/915) or only mild clinical signs for single days (R/278). Moreover, for R/824 (group A) clinical symptoms started as late as 19 dpi, and the clinical score increased rapidly (Figure 1 and Figure 3). One possible explanation is a prolonged incubation time, compared to the other experimentally infected animals. For natural outbreaks, incubation periods of up to four weeks are described [7,13]. Taking into consideration that LSDV is mainly transmitted mechanically via arthropod vectors [20,21,22,23], and, therefore, the number of infectious particles transmitted by the vector might be very low, it is possible that R/824 reacted similarly to naturally infected cattle. On the other hand, transmission from a severely affected to naïve animals also has been observed when the animals shared a drinking trough [13]. Given the fact that R/280 of the same group as R/824 developed generalized LSD (Figure 1, Figure 3) and viral shedding could be observed for R/280 in the days before euthanasia (Table 2 group A, Figure 3), and considering that R/824 did not seroconvert after inoculation (Table 4, Figure 3), virus transmission, e.g., via the watering system or the common feeding trough or direct contact, cannot be fully excluded. The clinical reactions of cattle of group B, C and D are similar. Here, some animals showed only mild to moderate clinical signs, whereas others were severely affected by LSDV and had to be euthanized before the end of the animal trial (Figure 1, Figure 3). Compared to the finding of our previous study also using LSDV-“Macedonia2016”, the overall patterns of the clinical scores of group B-D are similar to the clinical reaction observed in our last study. Additionally, the number of severely affected animals that were euthanized during the trial is comparable to that observed recently for LSDV-“Macedonia2016” [28]. However, two animals of group B (R/857, R/891) showed clearly lower clinical scores after inoculation compared to all other cattle of group B–D while displaying only slight nasal discharge and reduced food intake or activity for a few days (Figure 1, Figure 3). Furthermore, for both of them no increase of body temperatures could be observed during the whole study. Since it is known that there are non-febrile cases of LSDV [7], this finding is not very surprising. Due to the molecular data showing viremia for animal R/891 and viral genome in the nasal swabs of both animals (Table 2 group B, Figure 3), these data indicate a sub-clinical to mild clinical course of LSD for these two animals. Regarding the overall molecular analyses, no marked differences between the four inoculation groups could be seen (Table 2 and Table 3, Figure 3). In each group, also in the group with the lowest virus titer, viremia, as well as positive nasal swabs, were detected. However, viremia, as well as shedding of virus, seemed to be slightly reduced in the mild and moderately affected cattle of group A compared to the cattle of the other inoculation groups. Furthermore, generalized diseased cattle of all groups showed higher viral genome loads in EDTA blood and serum, indicating stronger viremia, as well as lower Cq values, in the nasal swabs compared to cattle displaying milder forms of LSD (Table 2). Similar findings were also seen in the study of Carn and Kitching after experimental infection of cattle with a virulent LSDV-Neethling strain [12]. Seroconversion started around 28 dpi in group B and C, whereas the first animals of group D scored positive in the SNT at 17 dpi (Table 4, Figure 3), which fits nicely with previous reports. After vaccination using live attenuated vaccines against LSDV or infection with virulent LSDV strains, first antibody detection is described at 10 dpi [16], around 14/15 dpi [17,28], and 21 dpi [11], respectively. In group B–D, all animals that survived until the end of the trial showed positive results in the ELISA, as well as in the SNT, at least starting at 28 dpi, except animal R/857 (group B). In contrast, no serological response could be detected for all cattle of group A (Table 4, Figure 3). This observation might be explained by the finding that no direct correlation between the amount of neutralizing antibodies and the immune status of a previously infected or vaccinated animal has been observed [16,17]. Tuppurainen et al. postulate that antibody levels of vaccinated or animals with mild LSD infection often might be below the detection limit of currently used serological tests, although these animals would be protected against a challenge infection [17].

In conclusion, our data indicate that intravenous inoculation of approximately 400 infectious LSDV particles of LSDV-“Macedonia2016” is sufficient to successfully infect cattle with LSDV and is even able to produce generalized LSD (Figure 1, Figure 3). This finding correlates well with the results of Carn and Kitching in 1995, that a titer of 10^3.3^ TCID_50_/_mL_ of a virulent LSDV-Neethling strain from South Africa is able to cause generalized LSDV infections when inoculated intravenously [12]. Moreover, in their study, a dose response regarding the severity of the disease could not be seen [12], which is similar to our results, to a certain extent. For the three groups infected with higher infectious titers than 10^2^ CCID_50_, we also did not see any correlation between the infective dose and the severity of clinical symptoms. Nevertheless, the severity of the clinical signs of the non-generalized cattle of group A (10^2^ CCID_50_) seemed to be slightly reduced compared to group B–D (Figure 1, Figure 3). The nearly missing correlation between inoculation dose and manifestation of clinical disease is a highly interesting point. In the field, morbidity varies between 3–85% [14,33], which is mainly influenced by the immune status of the hosts and the abundance of mechanical arthropod vectors [7,34]. However, detailed information and studies dealing with these findings are missing until now. Since we diluted the viral stock in the same medium that was also used for propagation of virus in cell culture, we assume that there is no impact of substances in the inoculum having impact on infectivity of the used virus stock or resistance of the cattle. Based on all these results, we conclude that the intravenous inoculation of 10^5^ to 10^6^ CCID_50_ of LSDV-“Macedonia2016” in cattle provides a robust challenge model for LSDV in further pathogenesis or vaccine efficacy studies.

## Figures and Tables

**Figure 1 viruses-12-00768-f001:**
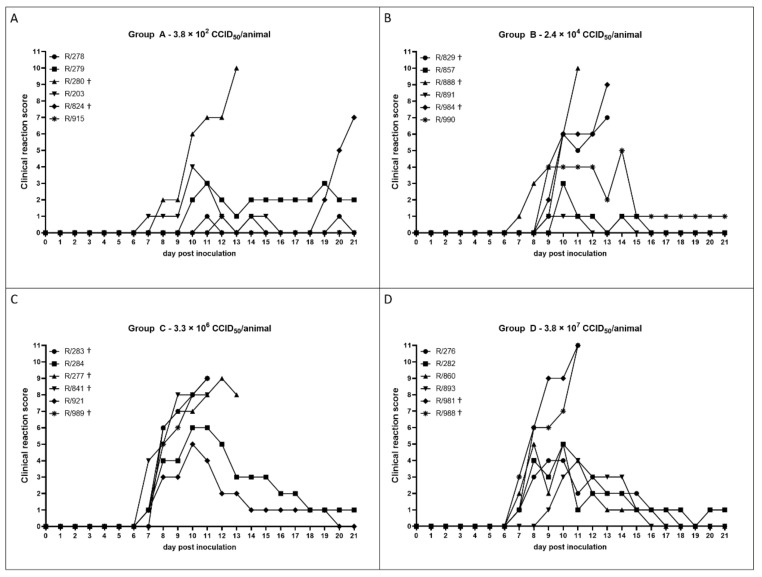
Clinical reaction score of cattle inoculated with different amounts of infectious *Lumpy Skin Disease virus* (LSDV)-“Macedonia2016”. Clinical reaction score (CRS) was measured daily from 0 dpi until 21 dpi for the different inoculation groups. CRS = 0 means no clinical signs, CRS 1–5 displays mild clinical course, and CRS 6–10 shows severe clinical reaction. Cattle with a clinical score of ≥10 were euthanized due to ethical reasons. (**A**) CRS of cattle inoculated with 3.8 × 10^2^ CCID_50_/animal. (**B**) CRS of cattle inoculated with 2.4 × 10^4^ CCID_50_/animal. (**C**) CRS of cattle inoculated with 3.3 × 10^6^ CCID_50_/animal. (**D**) CRS of cattle inoculated with 3.8 × 10^7^ CCID_50_/animal. † displays euthanasia of the respective animal before the end of the study.

**Figure 2 viruses-12-00768-f002:**
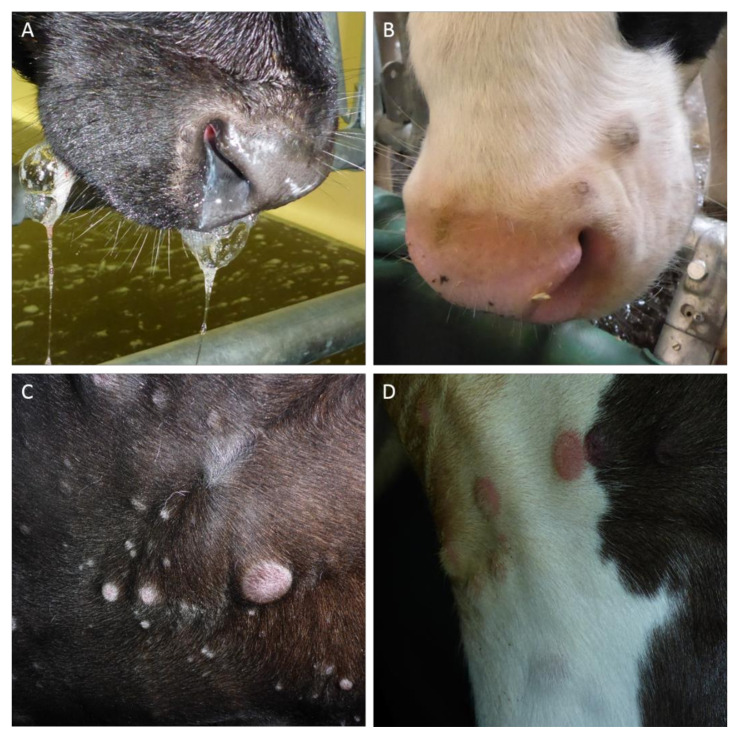
Clinical reactions of cattle following experimental infection with LSDV-strain “Macedonia2016”. In cattle with severe clinical signs, excessive salivation and nasal discharge (**A**), as well as sporadic to generalized skin nodules (**B**–**D**), could be observed.

**Figure 3 viruses-12-00768-f003:**
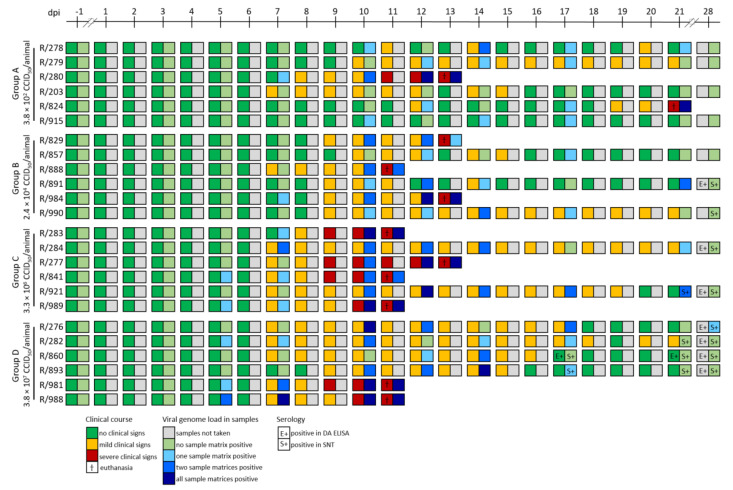
Summary graph of clinical, molecular and serological data. Clinical scores and body temperatures were measured daily during the animal trial. At certain days post infection, Ethylenediaminetetraacetic acid (EDTA) blood, serum, as well as nasal swab samples were collected and analyzed regarding their viral genome load. Moreover, serological tests were performed using both a double antigen (DA) an Enzyme-Linked Immunosorbent Assay (ELISA) and the serum neutralization test (SNT).

**Table 1 viruses-12-00768-t001:** Clinical reaction score system used during the described animal trial.

Evaluation Group	Symptomatic	Score
**General Condition**	Unremarkable	0
	Reduced activity	1
	Reduced activity, depressive	2
	Increased resting, clearly reduced activity	3
	Apathy and/or lateral position, animal does not react to stimuli	Abandonment
**Feed and Water Intake**	Unremarkable	0
	Reduced feed or water intake, increased salivation	1
	Clearly reduced feed or water intake	2
	No feed or water intake (1 day)	3
	No feed or water intake (2 day)	Abandonment
**Respiratory Tract**	Unremarkable	0
	Slight cough or nasal discharge	1
	Strong cough or nasal discharge combined with slight respiratory problems	2
	Strong cough or nasal discharge combined with moderate respiratory problems	3
	Strong cough or nasal discharge combined with strong respiratory problems	Abandonment
**Skin**	Unremarkable	0
	Sporadic skin nodules/lesions/papules	1
	Many skin nodules/lesions/papules	2
	Generalized skin lesions/nodules/papules	3
	Multiple abscesses or phlegmons	Abandonment
**Other Clinical Signs**	Ocular discharge	1
	Sporadic swelling of lymph nodes	1
	Multiple swelling of lymph nodes	2

**Table 2 viruses-12-00768-t002:** Pan Capripox real-time qPCR: Results of different sample materials after inoculation with different infective doses of LSDV-strain “Macedonia2016”.

	Cq-Values at Different Days Post Inoculation										
Group A 3.8 × 10^2^ CCID_50_/animal	−1/3	5	7	10	11	12	13	14	17	21	28
R/279_EDTA blood	no Cq	no Cq	no Cq	no Cq	-	no Cq	-	no Cq	no Cq	no Cq	no Cq
R/279_serum	no Cq	no Cq	no Cq	no Cq	-	38.6	-	no Cq	no Cq	no Cq	no Cq
R/279_nasal swab	no Cq	no Cq	no Cq	no Cq	-	no Cq	-	38.2	38.7	no Cq	no Cq
R/278_EDTA blood	no Cq	no Cq	no Cq	38.0	-	no Cq	-	29.6	no Cq	38.5	no Cq
R/278_serum	no Cq	no Cq	no Cq	no Cq	-	no Cq	-	33.6	no Cq	no Cq	no Cq
R/278_nasal swab	no Cq	no Cq	no Cq	no Cq	-	no Cq	-	no Cq	38.8	no Cq	no Cq
R/280_EDTA blood	no Cq	no Cq	no Cq	35.8	-	31.0	36.1	†	†	†	†
R/280_serum	no Cq	no Cq	37.1	38.3	-	34.9	36.0	†	†	†	†
R/280_nasal swab	no Cq	no Cq	no Cq	no Cq	-	34.8	29.1	†	†	†	†
R/824_EDTA blood	no Cq	no Cq	no Cq	no Cq	-	no Cq	-	no Cq	no Cq	30.7	†
R/824_serum	no Cq	no Cq	no Cq	no Cq	-	no Cq	-	no Cq	no Cq	36.1	†
R/824_nasal swab	no Cq	no Cq	no Cq	no Cq	-	37.3	-	no Cq	37.5	35.1	†
R/915_EDTA blood	no Cq	no Cq	no Cq	no Cq	-	no Cq	-	no Cq	36.8	no Cq	no Cq
R/915_serum	no Cq	no Cq	no Cq	37.5	-	no Cq	-	37.5	no Cq	no Cq	no Cq
R/915_nasal swab	no Cq	no Cq	no Cq	no Cq	-	no Cq	-	no Cq	no Cq	no Cq	no Cq
R/203_EDTA blood	no Cq	no Cq	no Cq	no Cq	-	no Cq	-	no Cq	no Cq	no Cq	no Cq
R/203_serum	no Cq	no Cq	no Cq	no Cq	-	no Cq	-	no Cq	no Cq	no Cq	no Cq
R/203_nasal swab	no Cq	no Cq	no Cq	no Cq	-	no Cq	-	no Cq	no Cq	no Cq	no Cq
**Group B 2.4 × 10^4^ CCID_50_/animal**											
R/888_EDTA blood	no Cq	no Cq	no Cq	32.3	37.5	†	†	†	†	†	†
R/888_serum	no Cq	no Cq	no Cq	34.6	36.2	†	†	†	†	†	†
R/888_nasal swab	no Cq	-	no Cq	no Cq	no Cq	†	†	†	†	†	†
R/857_EDTA blood	no Cq	no Cq	no Cq	no Cq	-	no Cq	-	no Cq	no Cq	no Cq	no Cq
R/857_serum	no Cq	no Cq	no Cq	no Cq	-	no Cq	-	no Cq	no Cq	no Cq	no Cq
R/857_nasal swab	no Cq	no Cq	no Cq	no Cq	-	38.3	-	no Cq	36.9	no Cq	no Cq
R/891_EDTA blood	no Cq	no Cq	no Cq	28.4	-	37.7	-	38.5	no Cq	35.8	no Cq
R/891_serum	no Cq	no Cq	no Cq	no Cq	-	no Cq	-	no Cq	no Cq	36.2	no Cq
R/891_nasal swab	no Cq	no Cq	no Cq	no Cq	-	38.9	-	no Cq	no Cq	no Cq	no Cq
R/829_EDTA blood	no Cq	no Cq	no Cq	31.9	-	38.3	38.5	†	†	†	†
R/829_serum	no Cq	no Cq	no Cq	37.2	-	38.0	no Cq	†	†	†	†
R/829_nasal swab	no Cq	no Cq	no Cq	no Cq	-	no Cq	no Cq	†	†	†	†
R/984_EDTA blood	no Cq	no Cq	37.4	33.7	-	31.1	30.9	†	†	†	†
R/984_serum	no Cq	no Cq	no Cq	35.7	-	36.3	33.5	†	†	†	†
R/984_nasal swab	no Cq	no Cq	no Cq	no Cq	-	31.2	34.2	†	†	†	†
R/990_EDTA blood	no Cq	no Cq	no Cq	37.3	-	40.9	-	38.9	no Cq	no Cq	no Cq
R/990_serum	no Cq	no Cq	no Cq	no Cq	-	no Cq	-	no Cq	no Cq	no Cq	no Cq
R/990_nasal swab	no Cq	no Cq	no Cq	no Cq	-	no Cq	-	37.4	38.0	no Cq	no Cq
**Group C 3.3 × 10^6^ CCID_50_/animal**											
R/277_EDTA blood	no Cq	no Cq	no Cq	34.2	-	36.8	35.4	†	†	†	†
R/277_serum	no Cq	no Cq	no Cq	37.7	-	37.1	37.4	†	†	†	†
R/277_nasal swab	no Cq	no Cq	no Cq	no Cq	-	38.3	37.6	†	†	†	†
R/283_EDTA blood	no Cq	no Cq	36.7	28.3	29.4	†	†	†	†	†	†
R/283_serum	no Cq	no Cq	no Cq	35.1	35.6	†	†	†	†	†	†
R/283_nasal swab	no Cq	no Cq	no Cq	35.0	34.5	†	†	†	†	†	†
R/284_EDTA blood	no Cq	no Cq	31.8	33.0	-	33.0	-	36.8	no Cq	no Cq	no Cq
R/284_serum	no Cq	no Cq	37.6	35.4	-	34.9	-	38.4	no Cq	37.6	no Cq
R/284_nasal swab	no Cq	no Cq	no Cq	no Cq	-	no Cq	-	no Cq	no Cq	no Cq	no Cq
R/921_EDTA blood	no Cq	no Cq	no Cq	28.2	-	32.6	-	33.3	32.6	35.3	no Cq
R/921_serum	no Cq	no Cq	no Cq	34.6	-	34.3	-	34.6	35.5	37.1	no Cq
R/921_nasal swab	no Cq	no Cq	no Cq	no Cq	-	38.5	-	no Cq	no Cq	no Cq	no Cq
R/841_EDTA blood	no Cq	34.8	34.3	34.8	33.1	†	†	†	†	†	†
R/841_serum	no Cq	no Cq	no Cq	35.5	35.7	†	†	†	†	†	†
R/841_nasal swab	no Cq	no Cq	no Cq	no Cq	no Cq	†	†	†	†	†	†
R/989_EDTA blood	no Cq	no Cq	33.8	29.6	34.6	†	†	†	†	†	†
R/989_serum	no Cq	37.3	no Cq	33.1	35.0	†	†	†	†	†	†
R/989_nasal swab	no Cq	no Cq	no Cq	33.7	28.5	†	†	†	†	†	†
**Group D 3.8 × 10^7^ CCID_50_/animal**											
R/276_EDTA blood	no Cq	no Cq	no Cq	24.3	-	38.8	-	no Cq	31.0	no Cq	30.9
R/276_serum	no Cq	no Cq	no Cq	33.0	-	36.9	-	no Cq	38.4	no Cq	no Cq
R/276_nasal swab	no Cq	no Cq	no Cq	37.2	-	no Cq	-	no Cq	no Cq	no Cq	no Cq
R/282_EDTA blood	no Cq	36.5	no Cq	38.5	-	no Cq	-	no Cq	no Cq	no Cq	no Cq
R/282_serum	no Cq	no Cq	38.6	no Cq	-	no Cq	-	no Cq	no Cq	no Cq	no Cq
R/282_nasal swab	no Cq	no Cq	no Cq	36.4	-	no Cq	-	34.7	37.0	no Cq	no Cq
R/981_EDTA blood	no Cq	31.7	33.9	32.4	30.4	†	†	†	†	†	†
R/981_serum	no Cq	no Cq	36.7	34.4	34.0	†	†	†	†	†	†
R/981_nasal swab	no Cq	no Cq	no Cq	26.8	24.6	†	†	†	†	†	†
R/860_EDTA blood	no Cq	no Cq	no Cq	no Cq	-	35.7	-	38.6	no Cq	no Cq	no Cq
R/860_serum	no Cq	no Cq	no Cq	no Cq	-	no Cq	-	39.4	no Cq	no Cq	no Cq
R/860_nasal swab	no Cq	no Cq	no Cq	no Cq	-	no Cq	-	no Cq	no Cq	no Cq	no Cq
R/893_EDTA blood	no Cq	no Cq	no Cq	33.8	-	36.1	-	36.3	no Cq	no Cq	no Cq
R/893_serum	no Cq	no Cq	no Cq	37.6	-	no Cq	-	37.2	no Cq	no Cq	no Cq
R/893_nasal swab	no Cq	no Cq	no Cq	no Cq	-	33.9	-	38.5	40.6	no Cq	no Cq
R/988_EDTA blood	no Cq	33.5	29.5	27.0	27.8	†	†	†	†	†	†
R/988_serum	no Cq	37.5	33.7	29.8	28.2	†	†	†	†	†	†
R/988_nasal swab	no Cq	no Cq	36.2	27.6	23.2	†	†	†	†	†	†

† displays animal was euthanized before sampling day.

**Table 3 viruses-12-00768-t003:** Pan Capripox real-time qPCR results of different organ samples of cattle after inoculation with different infective doses of LSDV-“Macedonia2016”.

Cattle		Cervical Lymph Node	Medestinal Lymph Node	Mesenterial Lymph Node	Spleen	Liver	Lung	Neck (Skin Nodules)	Udder Mirror (Skin Nodules)	Pox-Like Lesions Lung
Group A—3.8 × 10^2^ CCID_50_/animal	R/279	no Cq	no Cq	no Cq	no Cq	no Cq	no Cq	no Cq	n.t.	n.t.
	R/278	no Cq	no Cq	no Cq	no Cq	no Cq	no Cq	n.t.	n.t.	n.t.
	R/280	29.8	no Cq	no Cq	no Cq	38.4	37.7	18.6	24.0	n.t.
	R/824	29.3	no Cq	no Cq	35.7	37.2	30.2	18.9	17.1	15.9
	R/915	36.7	no Cq	no Cq	no Cq	no Cq	no Cq	n.t.	n.t.	n.t.
	R/203	no Cq	no Cq	no Cq	no Cq	no Cq	no Cq	n.t.	n.t.	n.t.
Group B—2.4 × 10^4^ CCID_50_/animal	R/888	32.4	no Cq	no Cq	no Cq	no Cq	35.4	19.3	21.2	n.t.
	R/857	no Cq	no Cq	no Cq	no Cq	no Cq	no Cq	n.t.	n.t.	n.t.
	R/891	no Cq	no Cq	no Cq	no Cq	no Cq	no Cq	n.t.	n.t.	n.t.
	R/829	36.4	no Cq	no Cq	no Cq	no Cq	no Cq	n.t.	n.t.	n.t.
	R/984	33.3	no Cq	no Cq	no Cq	37.6	33.9	21.8	20.8	n.t.
	R/990	no Cq	no Cq	no Cq	no Cq	no Cq	no Cq	n.t.	n.t.	n.t.
Group C—3.3 × 10^6^ CCID_50_/animal	R/277	no Cq	no Cq	no Cq	no Cq	no Cq	34.2	16.3	14.7	21.3
	R/283	31.0	37.5	37.1	no Cq	no Cq	33.5	19.7	17.4	n.t.
	R/284	no Cq	no Cq	no Cq	no Cq	no Cq	no Cq	no Cq	n.t.	23.1
	R/921	no Cq	no Cq	no Cq	no Cq	no Cq	no Cq	n.t.	n.t.	n.t.
	R/841	31.2	34.8	no Cq	no Cq	no Cq	33.3	20.0	19.7	n.t.
	R/989	31.8	31.3	no Cq	no Cq	no Cq	35.6	20.1	n.t.	23.8
Group D—3.8 × 10^7^ CCID_50_/animal	R/276	no Cq	no Cq	no Cq	no Cq	no Cq	no Cq	n.t.	n.t.	23.9
	R/282	no Cq	no Cq	no Cq	no Cq	no Cq	no Cq	n.t.	n.t.	n.t.
	R/981	31.8	41.1	no Cq	no Cq	no Cq	31.0	20.7	17.9	n.t.
	R/860	no Cq	no Cq	no Cq	no Cq	no Cq	31.3	n.t.	n.t.	n.t.
	R/893	no Cq	no Cq	no Cq	no Cq	no Cq	no Cq	no Cq	no Cq	25.8
	R/988	25.1	20.5	35.5	26.0	29.7	19.3	19.3	17.4	15.6

n.t. displays sample not taken.

**Table 4 viruses-12-00768-t004:** Serological results of certain serum samples taken during the animal trial using the ID Screen Capripox Double Antigen ELISA (ID.vet), as well as the serum neutralization test (SNT).

Group A 3.8 × 10^2^ CCID_50_/Animal			Group B 2.4 × 10^4^ CCID_50_/Animal			Group C 3.3 × 10^6^ CCID_50_/Animal			Group D 3.8 × 10^7^ CCID_50_/Animal		
Cattle _Dpi	ELISA S/P%	SNT Titer	Cattle_Dpi	ELISA S/P%	SNT Titer	Cattle_Dpi	ELISA S/P%	SNT Titer	Cattle_Dpi	ELISA S/P%	SNT Titer
R/279_−1	−1	n.d.	R/888_−1	0	n.d.	R/277_−1	−1	n.d.	R/276_−1	−1	n.d.
R/279_14	0	n.d.	R/888_11	−1	<1:10	R/277_13	5	<1:10	R/276_14	8	n.d.
R/279_17	−1	n.d.							R/276_17	11	n.d.
R/279_21	0	<1:10							R/276_21	11	1:13
R/279_28	0	<1:10							R/276_28	41	1:160
R/278_−1	0	n.d.	R/857_−1	1	n.d.	R/283_−1	−1	n.d.	R/282_−1	−1	n.d.
R/278_14	0	n.d.	R/857_14	0	n.d.	R/283_11	−1	<1:10	R/282_14	4	n.d.
R/278_17	0	n.d.	R/857_17	0	n.d.				R/282_17	14	1:16
R/278_21	−1	<1:10	R/857_21	1	<1:10				R/282_21	22	1:32
R/278_28	3	<1:10	R/857_28	6	≤1:13				R/282_28	43	1:128
R/280_−1	−2	n.d.	R/891_−1	0	n.d.	R/284_−1	−1	n.d.	R/981_−1	8	n.d.
R/280_13	1	<1:10	R/891_14	−1	n.d.	R/284_14	11	n.d.	R/981_11	14	<1:10
			R/891_17	1	n.d.	R/284_17	11	n.d.			
			R/891_21	0	<1:10	R/284_21	4	1:16			
			R/891_28	128	1:40	R/284_28	57	1:50			
R/824_−1	0	n.d.	R/829_−1	−1	n.d.	R/921_−1	−1	n.d.	R/860_−1	1	n.d.
R/824_14	3	n.d.	R/829_13	8	<1:10	R/921_14	4	<1:10	R/860_14	26	1:16
R/824_17	0	<1:10				R/921_17	13	<1:10	R/860_17	59	1:128
R/824_21	−1	<1:10				R/921_21	25	1:32	R/860_21	98	1:128
						R/921_28	67	1:80	R/860_28	163	1:256
R/915_−1	−1	n.d.	R/984_−1	−1	n.d.	R/841_−1	2	n.d.	R/893_−1	12	n.d.
R/915_14	−1	n.d.	R/984_13	5	<1:10	R/841_11	0	<1:10	R/893_14	7	<1:10
R/915_17	−1	n.d.							R/893_17	16	1:50
R/915_21	−2	<1:10							R/893_21	9	1:80
R/915_28	3	≤1:13							R/893_28	36	1:50
R/203_−1	−1	n.d.	R/990_−1	2	n.d.	R/989_−1	2	n.d.	R/988_−1	1	n.d.
R/203_14	0	n.d.	R/990_14	6	n.d.	R/989_11	2	<1:10	R/988_11	8	<1:10
R/203_17	0	n.d.	R/990_17	0	n.d.						
R/203_21	−1	<1:10	R/990_21	7	<1:10						
R/203_28	2	<1:10	R/990_28	3	1:20						

An S/P% ratio ≥ 30 was defined as positive in the DA ELISA, whereas a neutralizing titer of ≥ 1:20 in the SNT displays a positive result; n.d. means not determined.

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
