# Peer review of "Minimum Infective Dose of a Lumpy Skin Disease Virus Field Strain from North Macedonia"

_viruses, 2020, doi:10.3390/v12070768_

Round 1

Reviewer 1 Report

Wolff et al have conducted a careful study of the clinical course of infection of cattle with Lumpy skin disease virus after intravenous inoculation. In addition to observation of disease symptoms they have followed viremia and virus shedding by qPCR as well as seroconversion using ELISA and sero-neutralisation. The large amount of data obtained is presented in a logical and clear fashion. The major variable investigated was the dose of virus able to induce disease and various signs of infection including virus load and the immune response. Quite surprisingly there appears to be a very poor correlation between the viral dose used to infect the animals and clinical signs of infection and this conclusion is in agreement with findings in the literature referenced by Wolff et al. This poor correlation would require a little bit more discussion in the manuscript. Could there be some heterogeneity in the cattle that might confound the results with some animals being intrinsically more resistant than others? If this were the case then it would probably be necessary to carry out a study on a much larger cohort of animals, probably unrealistic for economic and practical reasons. Could the inoculum contain in addition to infectious virus, material that conveys resistance to infection. Note that the authors provide a reference for the way the infectious material was prepared but it would be preferable that they actually describe preparation of the inoculum in this manuscript.  

Minor points:

Line 112 “due” should be “according to”

Table 2 could be simplified by removing days -1 and 3 for which there are no Cq values. It would also be useful to add a legend for table 2. Furthermore, if possible please provide genome copy numbers instead of Cq values since they are more readily interpretable.         

Author Response

  • Wolff et al have conducted a careful study of the clinical course of infection of cattle with Lumpy skin disease virus after intravenous inoculation. In addition to observation of disease symptoms they have followed viremia and virus shedding by qPCR as well as seroconversion using ELISA and sero-neutralisation. The large amount of data obtained is presented in a logical and clear fashion. The major variable investigated was the dose of virus able to induce disease and various signs of infection including virus load and the immune response. Quite surprisingly there appears to be a very poor correlation between the viral dose used to infect the animals and clinical signs of infection and this conclusion is in agreement with findings in the literature referenced by Wolff et al. This poor correlation would require a little bit more discussion in the manuscript. Could there be some heterogeneity in the cattle that might confound the results with some animals being intrinsically more resistant than others? If this were the case then it would probably be necessary to carry out a study on a much larger cohort of animals, probably unrealistic for economic and practical reasons. Could the inoculum contain in addition to infectious virus, material that conveys resistance to infection. Note that the authors provide a reference for the way the infectious material was prepared but it would be preferable that they actually describe preparation of the inoculum in this manuscript.  

Answer: We thank the reviewer for the positive and detailed comments. The nearly missing correlation between inoculation dose and manifestation of clinical disease is a highly interesting point. In the field, morbidity varies between 3%-85%, which is mainly influenced by the immune status of the hosts and the abundance of mechanical arthropod vectors. However, detailed information and studies dealing with this findings are missing until now. Since we diluted the viral stock in the same medium that was also used for propagation of virus in cell culture, we assume that there is no impact of substances in the inoculum having impact on infectivity or resistance. We added a description about virus propagation and preparation of the inoculum in material and methods and discussed this point in the manuscript little more detailed.

Minor points:

  • Line 112 “due” should be “according to”

Answer: The respective phrase was corrected in the manuscript.

  • Table 2 could be simplified by removing days -1 and 3 for which there are no Cq values. It would also be useful to add a legend for table 2. Furthermore, if possible please provide genome copy numbers instead of Cq values since they are more readily interpretable.         

Answer: Since one Reviewer wanted the status before inoculation, we did not delete -1 dpi and 3 dpi from the table. Instead, we summarized -1 and 3 dpi in one single column.

For better comparison to other studies (especially in comparison to our previous study using LSDV-“Macedonia2016” for experimental infection of cattle), we used the semi-quantitative results of Cq values instead of number of genome copies for the manuscript. Additionally, we added genome copy numbers in two supplemental tables.

Reviewer 2 Report

In this paper, the goal is to detect the minimum infective dose able to induce a Lumpy skin disease in cattle using the Macedonia2016 strain. The clinical symptoms were studied for four infectious doses. Serological response and genome quantification in blood, serum and nasal swabs analyses were performed. Animals were euthanized under a 10 clinical reaction score. The study implies a lot of animal technical works to house 24 cattle under BSL4 conditions. Number of animals per group is relevant. This is a fastidious study.

Minor points

1-in materials and methods, it is noted that animal received 6 mL of virus suspension (line 90) while only 4 mL in the text (line 6 of discussion, “also shown that 4mL). this point has to be clarified.

2-line 91, clarify an “impact of the inoculation volume”. Can it be linked to a local virus-loading dose?

3-line 101, authors indicated that when cattle reached a clinical score under 10, the respective cattle was euthanized. In the study, several animals were euthanized even if their respective score was lower (1 for group A, 2 for group B, 4 for group C. The cut off (rule) of 10 could be modified because it is not representative of the reality. No description of death was done, and the number of dead animals is not linked to the dose. In this context the cause of death could be questioned, and their native health before study started.

4-the basal body temperature of cattle could be indicated and also the range of a body peak temperature. (in material and methods)

5-figure 1: to help the reader, a symbol (a cross) can be added closed to the last point of each animal which was euthanized, as some of them were euthanized with a clinical reaction score lower than 10.

6-In table 2, dead animals or euthanized animals are represented with a cross, this symbol could be added in both table 1 and table 3.

7-in figure 1, animal curve stopped at day 13 while in the text it is indicated line 166 that euthanasia occurred at day 11. Has to be clarified.

Author Response

  • In this paper, the goal is to detect the minimum infective dose able to induce a Lumpy skin disease in cattle using the Macedonia2016 strain. The clinical symptoms were studied for four infectious doses. Serological response and genome quantification in blood, serum and nasal swabs analyses were performed. Animals were euthanized under a 10 clinical reaction score. The study implies a lot of animal technical works to house 24 cattle under BSL4 conditions. Number of animals per group is relevant. This is a fastidious study.

Answer: We thank the reviewer for this positive evaluation of our study.

Minor points

  • 1-in materials and methods, it is noted that animal received 6 mL of virus suspension (line 90) while only 4 mL in the text (line 6 of discussion, “also shown that 4mL). this point has to be clarified.

Answer: In the study described in this manuscript, we used 6 ml of virus suspension to intravenously inoculate the respective cattle. In a previous study, we used 4 ml intravenously and subcutaneously (3 ml + 1 ml) of LSDV-“Macedonia2016” to experimentally infect cattle. We clarified this in the discussion.

  • 2-line 91, clarify an “impact of the inoculation volume”. Can it be linked to a local virus-loading dose?

Answer: Each animal should receive the same volume of inoculation material. This was thought to determine, in case of e.g. local reactions at the inoculation site, if these reactions were according to different viral loads and to exclude a possible impact of different volume of inoculation. We changed it in the text accordingly.

  • 3-line 101, authors indicated that when cattle reached a clinical score under 10, the respective cattle was euthanized. In the study, several animals were euthanized even if their respective score was lower (1 for group A, 2 for group B, 4 for group C. The cut off (rule) of 10 could be modified because it is not representative of the reality. No description of death was done, and the number of dead animals is not linked to the dose. In this context the cause of death could be questioned, and their native health before study started.

Answer: We thank the reviewer for this remark. Cattle were also euthanized when showing generalized LSD over a period of several days without showing clinical improvement, although if having a clinical reaction score <10. All animals were in good health before the start of the animal trial and no acute infections or poor health conditions could be observed. We included this information in material and methods.

  • 4-the basal body temperature of cattle could be indicated and also the range of a body peak temperature. (in material and methods)

Answer: Body temperature of the cattle previous to experimental infection was between 38.5°C and 39.5°C. We added this information in material and methods.

  • 5-figure 1: to help the reader, a symbol (a cross) can be added closed to the last point of each animal which was euthanized, as some of them were euthanized with a clinical reaction score lower than 10.

Answer: For better clarity, we added the symbol next to the animal ID in the legend of figure 1.

  • 6-In table 2, dead animals or euthanized animals are represented with a cross, this symbol could be added in both table 1 and table 3.

Answer: We included this symbol in the respective figures and tables.

  • 7-in figure 1, animal curve stopped at day 13 while in the text it is indicated line 166 that euthanasia occurred at day 11. Has to be clarified.

Answer: This was a mistake. Euthanasia at 13 dpi is correct as it is shown in figure 1. We changed it in the text accordingly.

Reviewer 3 Report

Major Problem:

Authors have not described the specific immune status of animals prior to virus infection. Description of serological response on page 14 refers to Table 4,which is missing. However, there are 2 Tables labelled as Table1. I presume one of these is Table 4, where authors have described ELISA and SNT results. None of the groups have been serologically tested on -1 dpi. Specific pre-immune status is an important consideration in describing the clinical picture of a disease caused by an infectious agent.

Minor issue:

Does LSDV form distinct plaques on cell monolayers, as other poxviruses? If so, pfu/ml should be defined in TCID or CCIF50. If no, TCID50/CCID50 is acceptable. But, it should be addressed in the discussion.

Author Response

Major Problem:

  • Authors have not described the specific immune status of animals prior to virus infection. Description of serological response on page 14 refers to Table 4,which is missing. However, there are 2 Tables labelled as Table1. I presume one of these is Table 4, where authors have described ELISA and SNT results. None of the groups have been serologically tested on -1 dpi. Specific pre-immune status is an important consideration in describing the clinical picture of a disease caused by an infectious agent.

Answer: We thank the reviewer for this remark. We changed the labels of the tables accordingly. The table displaying serological analyses of the cattle has to be table 4 as presumed by the reviewer. ELISA was performed at -1 dpi, representing the immune status of the animals before experimental inoculation. All animals were tested negative in the ELISA before inoculation with LSDV-“Macedonia2016”. We clarified this in the results.  

Minor issue:

  • Does LSDV form distinct plaques on cell monolayers, as other poxviruses? If so, pfu/ml should be defined in TCID or CCIF50. If no, TCID50/CCID50 is acceptable. But, it should be addressed in the discussion.

Answer: Most publications dealing with experimental Capripox virus infection use TCID50/CCID50. For determination of titer, we use titration and the Spearman-Kärber method. Plaque assay were not performed.

Round 2

Reviewer 3 Report

I noticed that authors have added a sentence regarding the health of the animals. I still think, it is not sufficient. Can authors give specific immunological evidence that these animals were not previously exposed to LSDV? It was my major issue with the manuscript.

Also minor language problems are added in the additions: These are mainly sentences starting with numbers.